# Elevated NET, Calprotectin, and Neopterin Levels Discriminate between Disease Activity in COVID-19, as Evidenced by Need for Hospitalization among Patients in Northern Italy

**DOI:** 10.3390/biomedicines12040766

**Published:** 2024-03-30

**Authors:** Geir Hetland, Magne Kristoffer Fagerhol, Mohammad Reza Mirlashari, Lise Sofie Haug Nissen-Meyer, Stefania Croci, Paola Adele Lonati, Martina Bonacini, Carlo Salvarani, Chiara Marvisi, Caterina Bodio, Francesco Muratore, Maria Orietta Borghi, Pier Luigi Meroni

**Affiliations:** 1Department of Immunology and Transfusion Medicine, Oslo University Hospital Ullevål, 0450 Oslo, Norway; geir.hetland@medisin.uio.no (G.H.); magnek@fagerhol.com (M.K.F.); mr.mirlashari@gmail.com (M.R.M.); lisoha@ous-hf.no (L.S.H.N.-M.); 2Department of Immunology, Institute of Clinical Medicine, University of Oslo, 0451 Oslo, Norway; 3Clinical Immunology, Allergy and Advanced Biotechnologies Unit, Azienda USL-IRCCS di Reggio Emilia, 42123 Reggio Emilia, Italy; stefania.croci@ausl.re.it (S.C.); martina.bonacini@ausl.re.it (M.B.); 4Research Laboratory of Immunorheumatology, IRCCS Istituto Auxologico Italiano, 20095 Cusano Milanino, Italy; p.lonati@auxologico.it (P.A.L.); c.bodio@auxologico.it (C.B.); or maria.borghi@unimi.it (M.O.B.); 5Azienda USL-IRCCS di Reggio Emilia e Università di Modena e Reggio Emilia, 42123 Reggio Emilia, Italy; carlo.salvarani@ausl.re.it (C.S.); chiara.marvisi@ausl.re.it (C.M.); francesco.muratore@ausl.re.it (F.M.); 6Department of Clinical Sciences and Community Health, University of Milan, 20122 Milan, Italy

**Keywords:** neutrophil extracellular traps, calprotectin, neopterin, complement, cytokines, COVID-19

## Abstract

Coronavirus disease 2019 (COVID-19) displays clinical heterogeneity, but little information is available for patients with mild or very early disease. We aimed to characterize biomarkers that are useful for discriminating the hospitalization risk in a COVID-19 cohort from Northern Italy during the first pandemic wave. We enrolled and followed for four weeks 76 symptomatic SARS-CoV-2 positive patients and age/sex-matched healthy controls. Patients with mild disease were discharged (n.42), and the remaining patients were hospitalized (n.34). Blood was collected before any anti-inflammatory/immunosuppressive therapy and assessed for soluble C5b-9/C5a, H3-neutrophil extracellular traps (NETs), calprotectin, and DNase plasma levels via ELISA and a panel of proinflammatory cytokines via ELLA. Calprotectin and NET levels discriminate between hospitalized and non-hospitalized patients, while DNase negatively correlates with NET levels; there are positive correlations between calprotectin and both NET and neopterin levels. Neopterin levels increase in patients at the beginning of the disease and do so more in hospitalized than non-hospitalized patients. C5a and sC5b-9, and other acute phase proteins, correlate with neopterin, calprotectin, and DNase. Both NET and neopterin levels negatively correlate with platelet count. We show that calprotectin, NETs, and neopterin are important proinflammatory parameters potentially useful for discriminating between COVID-19 patients at risk of hospitalization.

## 1. Introduction 

Coronavirus disease 2019 (COVID-19) is initially characterized by flu-like symptoms, which can be associated with pneumonia and multi-organ damage. The final clinical outcome varies greatly. While some patients may only experience a mild or moderate upper airway illness that does not require hospitalization, others may develop severe disease and life-threatening acute respiratory distress syndrome (ARDS). Older age and the presence of co-morbidities, such as diabetes mellitus, obesity, and immunosuppression, have been associated with more aggressive COVID-19 and higher mortality [1,2,3,4].

There is evidence that the more aggressive disease is largely caused by a hyperinflammatory and prothrombotic response (immunothrombosis) triggered by the SARS-CoV-2 virus and mediated by the host’s deregulated immune response [5,6]. Besides the increased production of proinflammatory cytokines, additional mediators of innate immunity, such as complement activation, are involved. In particular, the high levels of circulating complement activation products, together with systemic proinflammatory cytokines, are crucial in causing endothelial perturbation. Moreover, the tissue deposition of complement components may contribute to the induction of a pro-adhesive and thrombophilic endothelial phenotype as an amplification thrombo-inflammatory loop ultimately responsible for the thrombophilic state in COVID-19 [1]. 

Studies have shown that the hyperproduction of proinflammatory cytokines, the release of damage-associated molecular patterns during tissue injury, and an increased occurrence of thrombotic events are all associated with the involvement of neutrophils [7]. Specifically, the increased formation of low-density neutrophils and the generation of neutrophil extracellular traps (NETs) play significant roles in the immunopathology of the disease. They are closely correlated with its severity and poor prognosis [8,9,10]. An intense systemic inflammation with high levels of proinflammatory cytokines and the associated endothelial perturbation characterize SARS-CoV-2-infected patients who require hospitalization because of a more severe disease [11,12].

Calprotectin, formerly L1 protein, is the main cytosolic protein in neutrophils and monocytes [13]. Elevated levels of extracellular calprotectin are not only found in autoimmune diseases [14,15] but have also been detected in severe COVID-19, suggesting that neutrophils are involved in inflammation and respiratory exacerbation in COVID-19 [16]. Calprotectin is a major component of NETs, which are usually involved in host defense for the destruction of invading pathogens [17]. Increased NET levels have been found in hospitalized COVID-19 patients [10], where they can obstruct capillary circulation [18]. In severe COVID-19, NETs are considered to be a driver of endothelial damage for the ensuing immune thrombosis [19,20]. However, the formation of NET is counteracted by DNase, which is an enzyme that destroys NETs via the degradation of their DNA content [21]. In contrast to the previous and current findings in COVID-19 patients, we only detected moderately increased levels of calprotectin among the other parameters (NETs, neopterin, and syndecan-1) tested in COVID-19 convalescent blood donors in Oslo [22].

Additional studies have found that calprotectin and NET levels are raised in vaccine-induced thrombocytopenia and thrombosis (VITT) after vaccination with ChAdOx1 nCoV-19 [23,24,25]. Moreover, recently, we reported increased DNase levels in the ChAdOx1 nCoV-19-vaccinated individuals, as measured via a novel DNase test, which correlated with their previously measured NET values [26].

Neopterin is an anti-oxidant agent produced by mononuclear phagocytes upon stimulation with IFNγ, which is produced by T cells and NK cells when viruses, cancer cells, and other proinflammatory danger signals activate them [27]. IFN-related pattern recognition receptors (PRRs) are activated by pathogen-associated (PAMPs) and damage-associated (DAMPs) molecular patterns in the mucosa [28]. Previously, neopterin has been used as an unspecific screening test for virus infections in Austrian blood banks [29,30]. Elevated neopterin levels have also been found during SARS-CoV-2 infection and at higher levels in severe than mild disease [31].

The complement activation products C5a and sC5b-9 have been shown to increase rapidly in COVID-19 patients [32,33,34,35] and are predictors for the hospitalization of COVID-19 patients in Northern Italy [11]. Also, a range of other proinflammatory cytokines, growth factors, and hematological parameters increased during COVID-19, more so in severely affected patients during the first pandemic wave [11].

Here, we wanted to investigate whether the proinflammatory parameters, NETs, calprotectin, DNase, and neopterin could discriminate between hospitalized and non-hospitalized COVID-19 patients examined in the same patient cohort from Northern Italy.

Moreover, we examined whether there were correlations between neopterin, calprotectin, and NET levels and those previously measured for proinflammatory cytokines, complement activation products, and hematology parameters in this Northern Italian patient cohort.

## 2. Materials and Methods

### 2.1. Patients

We enrolled and prospectively followed up 76 symptomatic patients for suspected COVID-19 during the outbreak peak between 9 March 2020 and 22 April 2020. These patients were prospectively followed up. The inclusion criteria were an age > 18 years old and a diagnosis of COVID-19 confirmed via a positive RT-PCR for SARS-CoV-2 in at least one biological sample. Patients suffering from an infectious disease other than SARS-CoV-2, suffering from previous or current autoimmune diseases, and who were pregnant were excluded. Further clinical details have been reported previously [11].

A multidisciplinary team established a six-phenotype classification based on patient features, vital signs, medical histories, symptoms, blood test results, and instrumental findings to manage patients presenting to the emergency room (ER), as previously reported [36,37]. The instrumental diagnostic protocol for these patients included chest X-rays and CT scans in cases of positive X-ray findings and negative X-rays that had highly suggestive clinical features, respectively. Patients from the first two clinical phenotypes (phenotype 1 and 2A) were discharged and referred to the preventive medicine and public health department for follow-up. These patients usually displayed fever, with or without respiratory involvement and with or without radiological evidence of pneumonia (usually <20% of lung parenchymal involvement at CT-scan), and a negative walking test. All other patients were hospitalized. The hospitalized patients were significantly older than the non-hospitalized ones (mean age ± SD: 62.82 ± 11.05 vs. 52.86 ± 15.45; *p* < 0.005). The clinical outcome was evaluated after a 4-week follow-up period, and the patients were classified as having mild (n.42) or moderate–severe COVID-19 (n.34). Mild cases included patients with phenotypes 1 and 2A, who, during the follow-up, had stable disease and were not hospitalized; patients presenting with phenotypes 2B and 3 at ER admission and who did not worsen during the follow-up were considered to have moderate COVID-19. These patients were hospitalized and responded to conventional patients’ O_2_ therapy. Phenotypes 4 and 5 at ER admission or patients presenting with milder disease that worsened during the follow-up were considered severe cases. These patients were hospitalized and admitted to sub-intensive units or the intensive care unit dedicated to COVID-19 patients requiring non-invasive mechanical ventilation (NIV) or intubation.

On the day of admission or the following day, antecubital vein blood samples were collected in EDTA tubes for the measurement of sC5b-9 and C5a and in serum tubes for the measurement of the other parameters and stored at −80 °C until testing, as previously described [11]. Since Fragmin is used for treating most hospitalized COVID-19 patients, and this treatment destroys and lowers NET values, reliable NET testing could not be carried out in these patients [38,39]. This also applies to calprotectin, which is contained in NETs, and where the S100A9 (MRP-14) dimer part of calprotectin has an affinity for and interacts with heparin [38]. Therefore, these hospitalized COVID-19 patients were omitted from this study, in contrast to the original Northern Italian COVID-19 patient cohort included in the previous paper [11]. Moreover, DNase levels are reduced/hampered by EDTA, so their levels were lower in EDTA plasma than in corresponding/parallel serum samples from the patients and controls [26]. However, since the percentage reduction in DNase values must be the same in all EDTA samples, this would not affect comparisons of DNase values between groups or correlations with levels of other parameters in the same samples. All patients’ blood samples were collected before starting other treatments to avoid any drug interference with the biological parameters analyzed. All the hospitalized patients with moderate–severe COVID-19 were then treated with tocilizumab and/or glucocorticoids.

The results of the following laboratory parameters were obtained from the patients’ clinical records: fibrin fragment D-dimer, C-reactive protein (CRP), ferritin, white blood cells, neutrophils, lymphocytes, monocytes, platelets, prothrombin time (PT), activated partial thromboplastin time (aPTT), and fibrinogen [11]. The neutrophil/lymphocyte ratio (NLR) was also calculated.

This study was approved by the Area Vasta Emilia Nord (AVEN) Ethics Committee on 28 July 2020 (protocol number 855/2020/OSS/AUSLRE—COVID-2020-12371808) and carried out in conformity with the 2013 revision of the Declaration of Helsinki and the Code of Good Clinical Practice.

Control samples were from healthy Norwegian blood donors at Oslo Blood Bank, sampled in 2015 before the COVID-19 pandemic, except for the controls used for the in-neopterin test, who were healthy healthcare workers sampled in 2020/2021 (during the pandemic).

### 2.2. H3-NET Dual Hybrid ELISA

The assay was designed to detect complexes containing DNA and leucocyte calprotectin and performed as previously described [40]. In brief, 96-well microplates (ThermoFisher Scientific, Waltham, MA, USA) were coated with rabbit anti-histone 3 (H3) (Agrisera AB, Vännäs, Sweden) or anti-calprotectin (ThermoFisher Scientific, Waltham, MA, USA), the latter used to establish a standard curve for bound calprotectin in complexes with H3. Plasma samples were tested at 1:5 dilution. After incubation at room temperature (RT) for 90 min and three washes, the horse radish peroxidase (HRP)-conjugated monoclonal anti-calprotectin antibodies were added to the wells. The H3-NET amount was measured as 450 nm and expressed in ng/mL. 

### 2.3. Calprotectin Mixed Monoclonal Assay

A novel calprotectin ELISA based on a mixture of monoclonal antibodies (ProMab Corp, Richmond, CA, USA) was established to ensure that all calprotectin in biological materials containing both histone and DNA fragments could be reliably assayed. The monoclonals were selected to react with all chromatography fractions of stool extracts from inflammatory bowel syndrome patients. The mixed monoclonal (MiMo) antibodies were used both for the coating of microwells and preparation of a HRP conjugate [40]. In brief, wells were coated with 5 µg/mL MiMo in 0.1 M sodium citrate at pH 6 for two hours at RT. After washing, standards (recombinant calprotectin) and samples were adequately diluted and incubated via shaking for 40 min at RT, followed by washing and incubation with HRP-anti-MiMo. The calprotectin levels were measured at 450 nm and expressed in ng/mL. 

### 2.4. Competitive DNase ELISA

In brief, a competitive DNase assay was established using chicken antibodies (NABAS, Ås, Norway) as a coat and recombinant human DNase (Roche, Basel, Switzerland) that was conjugated with HRP and mixed with a sample to be examined for readout. Readings at 450 nm after the addition of substrate were inversely proportional to the DNase concentration in the sample, expressed in ng/mL [26].

### 2.5. Neopterin

Neopterin was measured using a commercial competitive ELISA (DEIA 1640; Creative Diagnostics^®^, Shirley, NY, USA) according to the manufacturer’s protocol. The neopterin concentration in the plasma samples was directly read from the standard curve. The results were reported as ng/mL. 

### 2.6. Complement Activation Products

The plasma levels of sC5b-9 and C5a were measured using solid-phase assays (MicroVue Complement SC5b-9 Plus EIA kit, MicroVue Complement C5a EIA, Quidel Corporation, San Diego, CA, USA), as previously described [11,34,35]. sC5b-9 displayed a 6.8% intra-assay coefficient of variation (CV) and 13.1% inter-assay CV; the lowest limit of detection was 3.7 ng/mL. Both intra- and inter-assay CVs were <12% for sC5a, whose lowest detection limit was 0.01 ng/mL. The assay cutoffs [95th percentile] were previously calculated by testing 50 healthy subjects matching patients’ sex and age selected from the archives of the IRCCS Isituto Auxologico Italiano and collected before the COVID-19 pandemic. The cutoff for sC5b-9 was 411.50 ng/mL, and the one for sC5a was 15.53 ng/mL. 

### 2.7. Cytokine Detection

We performed multi-analyte profiling of 17 soluble mediators in patients’ and controls’ serum samples using the automated microfluidic analyzer ELLA (BioTechne, Minneapolis, MN, USA) according to the manufacturer’s instructions [11]. The following mediators were analyzed: IL-1β, IL-6, IL-8, TNFα, IL-4, IL-10, IL-12p70, IFNγ, IFNα, VEGF-A, VEGF-B, GM-CSF, IL-2, IL-17A, VEGFR2, and BLyS. For measurement, the parameters were grouped through the required sample dilution into four panels. Factory-calibrated cartridges and built-in standard curves minimized inter-batch variability. Intra- and inter-assay CVs were <8%. The assay performance allowed a sub-picogram level of sensitivity. 

### 2.8. Statistics

One-way ANOVA was used to examine differences between several groups, and *t*-test/Wilcoxon was used to compare two groups. The Pearson correlation was applied to examine correlations between levels of different parameters in COVID-19 patients. *p* values < 0.05 were considered significant.

Statistical analysis was performed using GraphPad Prism, version 6.2. (GraphPad Software, Boston, MA, USA).

## 3. Results

### 3.1. Patients

Of the 76 patients included in this study, 42 were classified as having mild and 34 as having moderate–severe COVID-19 according to the scoring method detailed in the Section 2. Compared to non-hospitalized patients (mild), those who were hospitalized (moderate–severe) were older, were more likely to be males, and had greater lung parenchymal involvement, areas of ground-glass opacity, and parenchymal consolidation. 

### 3.2. Levels of NETs, Calprotectin, DNase, and Neopterin in Hospitalized versus Non-Hospitalized COVID-19 Patients

NET levels were increased 9-fold and 16-fold in non-hospitalized and hospitalized COVID-19 patients, respectively, compared with healthy blood donors (ANOVA *p* < 0.0001) (Figure 1A). Thus, NET levels were nearly two-fold higher in patients hospitalized for more severe COVID-19 than those not admitted to hospital for treatment (*p* < 0.0005).

The differences in calprotectin values were even more pronounced; the levels were over 4-fold and 14-fold elevated in non-hospitalized and hospitalized patients relative to blood donors (ANOVA *p* < 0.0001) (Figure 1B). This indicates 3-fold higher calprotectin levels in hospitalized than non-hospitalized COVID-19 patients (*p* < 0.0001).

DNase, which digests DNA in NET, was found to be at similar levels in all groups, with a slight tendency toward lower levels in the hospitalized patients but without any statistical significance (Figure 1C).

In fact, whereas there was a moderately positive correlation (r = 0.389, *p* = 0.0005) between levels of NETs and calprotectin in COVID-19 patients, a low negative correlation (r = −0.262, *p* = 0.022) was found between their NET and DNase levels (Table 1).

When neopterin was measured, there were 50% increased levels in hospitalized relative to non-hospitalized COVID-19 patients (*p* < 0.01), who, again, had nearly two-fold higher levels than healthy controls (*p* < 0.01) (Figure 2). There was a low-grade positive correlation between levels of calprotectin and neopterin in these COVID-19 patients (Table 1).

### 3.3. Correlations between NETs, Calprotectin, Neopterin, and the Other Parameters

Table 2 reports the correlations between NET, calprotectin, and neopterin levels on one side and levels of the complement activation products C5a, sC5b-9, proinflammatory cytokines, and inflammatory cells on the other. COVID-19 patients displayed a positive correlation between NET levels and NLR and a negative correlation with platelet count. There were positive correlations with sC5b-9 and leukocytes for calprotectin, which must also be true for neutrophils because of the negative correlations with lymphocytes and monocytes. There were also positive correlations between calprotectin and other parameters of systemic inflammation, such as ferritin, fibrinogen, and procalcitonin, or tissue damage (AST, LDH, CPK). It was found that DNase correlated negatively with some proinflammatory parameters (CRP, C5a, ferritin) and with BAFF levels but positively with IL-17A and VEGFR2. Neopterin correlated positively with proinflammatory biomarkers (CPR, procalcitonin, ferritin), BAFF and IFNα, and tissue damage markers (AST, LDH, CPK, and troponin HS) but negatively with albumin and platelets.

## 4. Discussion

The heterogeneity of the clinical outcome of the SARS-CoV-2 infection raises the need for parameters useful for identifying patients at higher risk of an aggressive disease who would require more conservative management. 

We confirmed, in our cohort, that older age is associated with a higher risk of hospitalization due to more severe disease, as previously reported [2], and we looked at additional biomarkers.

The increase in H3-NET levels in hospitalized versus non-hospitalized patients is consistent with the widely accepted involvement of neutrophil in COVID-19. The complement system is activated in COVID-19, and complement deposits can be found in damaged tissues such as the lungs, which have a chemotactic effect on neutrophils and ultimately favor NETosis [11,32,33,34,35,41]. We also showed increased levels of IL-8 in sera from the same cohort of patients in Northern Italy [11]. IL-8 may, in turn, cooperate in recruiting neutrophils to the inflammatory site, and there is evidence that high receptor-saturating IL-8 concentrations promote NETosis [42]. The resulting enhanced NETosis is associated with thrombosis, which leads to further complications at the center of the inflammatory insult [19,20].

Hence, the increase in calprotectin, mainly derived from neutrophils, is unsurprising in our COVID-19 cohort, in whom secondary pneumonia gives an influx of neutrophils to the lungs. It has been shown that calprotectin, which is a heterodimer of S100A8 (MRP-8) and S100A9 (MRP-14), is deposited on the endothelium of venules in inflamed tissue and also promotes the extravasation of leukocytes [38].

Neopterin is an old test for emerging infections [29,30], and it has also been documented for SARS-CoV-2 infection [31]. There is evidence that higher levels of neopterin in severe COVID-19 may be used as a predictive marker for disease severity [43,44]. This is confirmed here by the increased levels of neopterin and the correlation with other inflammatory markers in our COVID-19 cohort. Moreover, we find that neopterin concentrations can discriminate between hospitalized and non-hospitalized patients, similar to calprotectin and NET values. Previously, this has especially been shown for the complement activation products C5a and sC5b-9 and other proinflammatory markers [11].

Since heparin is known to be involved in NET formation/degradation and most COVID-19 hospitalized patients were treated with heparin (Fragmin), such patients could not be included when measuring NET levels [38,39]. Moreover, calprotectin cannot be measured accurately in patients treated with heparin because the S100A9 (MRP-14) dimer part of calprotectin has affinity and interacts with heparin [38]. Therefore, heparin-treated patients were excluded from the current paper.

The correlation between NET and NLR levels should be discussed because NLR is a parameter for physiological stress and particularly important with regard to lung affection and pneumonia during SARS-CoV-2 infection, where NETosis plays a critical role. The negative correlation between NETs and platelet count may indicate that neutrophil activation contributes to platelet activation, consumption, and/or involvement in the formation of leuco-aggregates [45].

There is a general agreement that SARS-CoV-2 plays a direct role in tissue damage, but the intense immune response further contributes to the clinical manifestations [5]. The relationships of NETs, calprotectin, and neopterin with COVID-19 hospitalization and, simultaneously, with the biomarkers of the innate and adaptive immune responses potentially responsible for tissue damage are consistent with this view.

DNase levels are comparable in controls and COVID-19, despite having slightly decreased values in hospitalized patients. Due to the negative correlation between DNase and many proinflammatory parameters, including CRP, we may speculate whether DNase has an anti-inflammatory effect. The negative correlation with complement activation product C5a, in particular, may suggest that DNase can inhibit C5 activation. On the other hand, the negative correlation of DNase with neutrophils is apparently surprising since these cells are the primary source of the target that DNase is supposed to digest. We could speculate that the increased release of neutrophil granule proteins associated with the DNA framework may engage a more significant amount of DNase that, in turn, is no longer available for the detection assay.

## 5. Conclusions

In conclusion, our report shows that a stronger innate immune response is linked to more severe COVID-19 clinical symptoms, in addition to an older age. In a previous study, we found that complement activation and the production of proinflammatory cytokines were associated with hospitalization in a COVID-19 patient group during the initial wave of the pandemic in Northern Italy. Our current findings indicate that elevated levels of NETs and calprotectin can differentiate disease activity and predict the need for hospitalization. This discovery aligns with our previous study and further supports the combined role of innate immunity in disease activity. Additionally, this study provides evidence and confirmation that neopterin is a valuable old test for screening emerging infections. 

It is worth noting that complement activation has also recently been reported in patients suffering from long-term COVID-19. This finding further supports the involvement of innate immunity not only in acute disease but also in chronic systemic inflammation triggered by SARS-CoV-2 infection [46].

Taken together, these biomarkers can help to identify COVID-19 patients who are at a higher risk of developing more severe disease. They can serve as useful predictive tools, along with older age and the presence of co-morbidities, which have been previously identified as risk factors for severe disease [47].

## Figures and Tables

**Figure 1 biomedicines-12-00766-f001:**
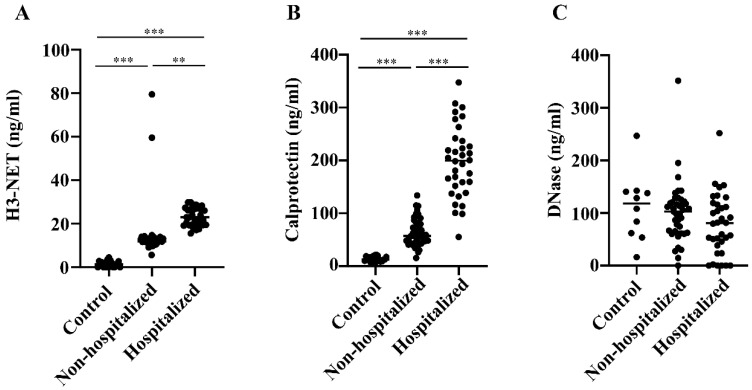
(**A**) NET levels in COVID-19 patients. Levels of Histone 3 (H3)-NETs were measured in plasma samples of Norwegian blood donors (n.20) and non-hospitalized (n.42) and hospitalized (n.34) COVID-19 patients from Northern Italy. (**B**) Calprotectin levels in COVID-19 patients. Levels of calprotectin were measured using a novel mixed monoclonal antibody assay in plasma samples of Norwegian blood donors (n.20) and non-hospitalized (n.42) and hospitalized (n.34) COVID-19 patients from Northern Italy. (**C**) DNase levels in COVID-19 patients. Levels of DNase were measured via a competitive ELISA in plasma samples of Norwegian blood donors (n.10) and non-hospitalized (n.42) and hospitalized (n.34) COVID-19 patients from Northern Italy. Differences between groups were assessed via one-way ANOVA and *t*-test/Wilcoxon. ** *p* < 0.0005 and *** *p* < 0.0001.

**Figure 2 biomedicines-12-00766-f002:**
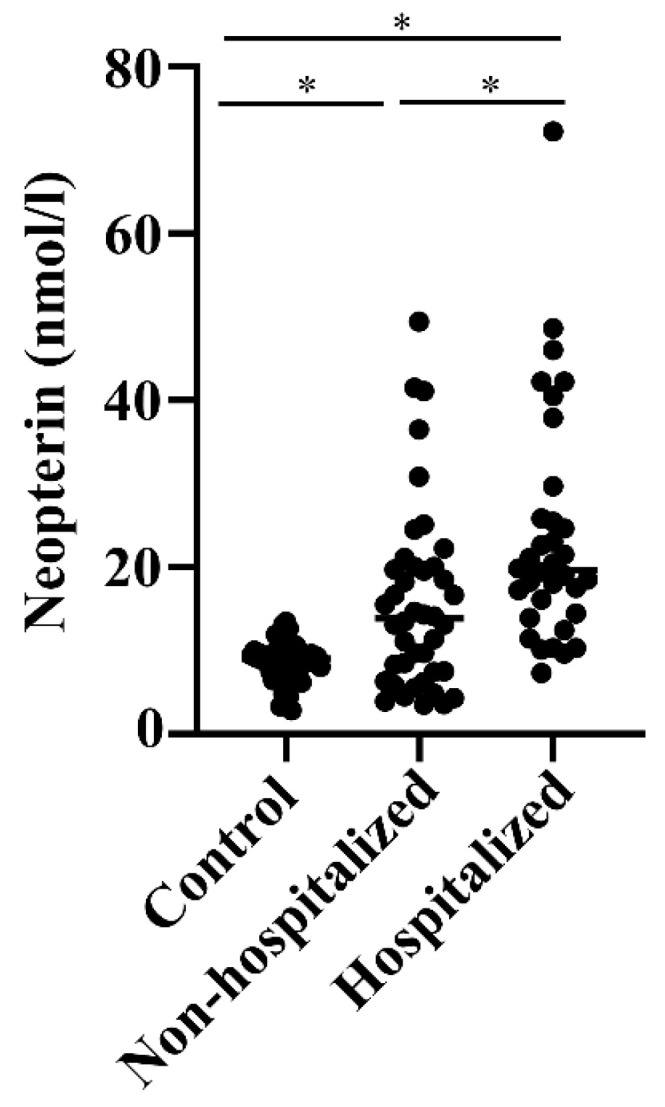
Neopterin levels in COVID-19 patients. Levels of neopterin were measured via a commercial assay in samples of healthy Norwegian health workers (n.53) and non-hospitalized (n.42) and hospitalized (n.34) COVID-19 patients from Northern Italy. Differences between groups were assessed via one-way ANOVA and *t*-test/Wilcoxon. * *p* < 0.01.

**Table 1 biomedicines-12-00766-t001:** NET, Calprotectin, DNase, and Neopterin levels in COVID-19 patients from Northern Italy.

Parameter	NETs	Calprotectin	DNase	Neopterin
NETs	1	r = 0.3894	r = −0.2628	r = 0.1939
		*p* = 0.0005	*p* = 0.0227	*p* = 0.0932
Calprotectin	-	1	r = −0.1703	r = 0.2752
			-	*p* = 0.0161
DNase	-	-	1	r = −0.1050
				-
Neopterin	-	-	-	1

The Pearson correlation was applied to examine correlations between levels of different parameters in COVID-19 patients. *p* values ≤ 0.05 were considered statistically significant.

**Table 2 biomedicines-12-00766-t002:** Correlations of neopterin, calprotectin, NET, and DNase levels with other parameters in COVID-19 patients from Northern Italy.

Parameter	Neopterin	Calprotectin	NET	DNase
CRP	0.3328 *	-	-	−0.4026 **
C5a	-	-	-	−0.3633 **
SC5b-9	-	0.2635 *	-	-
BAFF	0.5362 ****	0.2472 ^	-	−0.3422 **
IFNa	0.3480 *	0.3061 *	-	-
IL-17A	-	-	-	0.2619 *
VEGFR2	-	-	-	0.3595 **
Procalcitonin	0.5200 ***	0.6032 ***	-	-
Ferritin	0.5226 ***	0.3895 **	-	−0.3773 **
Fibrinogen	-	0.3895 *	-	-
AST	0.6641 ****	0.3562 **	-	−0.3228 *
LDH	0.5496 ****	0.4021 **	-	-
CPK	0.4146 **	0.4439 **	-	-
Troponin HS	0.6490 ****	-	-	-
Leukos	-	0.2500 ^	-	-
Lymphos	-	−0.3020 *	-	-
Monoc	-	−0.3835 **	-	-
NLR	-	-	0.3737 **	-
PLT	−0.4584 **	-	−0.2737 *	-

The Pearson correlation was applied to examine correlations between the levels of different parameters in COVID-19 patients. *p* values: ^ *p* = 0.05, * *p* < 0.05, ** *p* < 0.01, *** *p* < 0.001, and **** *p* < 0.0001.

## Data Availability

This set of raw data is accessible by request because it includes sensitive information. Please write your request to pierluigi.meroni@unimi.it.

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
