# Peer review of "Elevated NET, Calprotectin, and Neopterin Levels Discriminate between Disease Activity in COVID-19, as Evidenced by Need for Hospitalization among Patients in Northern Italy"

_biomedicines, 2024, doi:10.3390/biomedicines12040766_

Round 1

Reviewer 1 Report

Comments and Suggestions for Authors

Hetland describe important initial innate immune reactions against COVD 19 and correlate these findings to the length of hospitalization. These are invaluable information of a novel viral infection before vaccination.

In general, the data are well presented and put into the context.

1: It is understood that only data are shown that may statistically be evaluated. However important information that are present today may be lost tomorrow. One aspect that appears important is age of the patients as the immune system reacts clearly different in those individuals. Even if only tendencies can be described due to the limited number of individuals are described in this study, it would be important for future studies or emerging new viruses. The “garden” of coronaviruses is large. Even if these information may appear as a speculation, it will be important information of front line medical personal.

2: It may be interesting to make a further link of these data to long covid based on the recent publication by Cervia-Hasler et al 2024 Science

https://www.ncbi.nlm.nih.gov/pubmed/38236961

Author Response

We would like to thank the Referee for the constructive comments.

Hetland describe important initial innate immune reactions against COVD 19 and correlate these findings to the length of hospitalization. These are invaluable information of a novel viral infection before vaccination.

In general, the data are well presented and put into the context.

1: It is understood that only data are shown that may statistically be evaluated. However important information that are present today may be lost tomorrow. One aspect that appears important is age of the patients as the immune system reacts clearly different in those individuals. Even if only tendencies can be described due to the limited number of individuals are described in this study, it would be important for future studies or emerging new viruses. The “garden” of coronaviruses is large. Even if this information may appear as a speculation, it will be important information of front line medical personal.

We have included new information about the patients and data in the revised text. These new parts are highlighted throughout the text, and new references have been added.

2: It may be interesting to make a further link of these data to long covid based on the recent publication by Cervia-Hasler et al 2024 Science https://www.ncbi.nlm.nih.gov/pubmed/38236961

As suggested, we have discussed the paper on complement in long-COVID patients published in Science in the section "Conclusion", page 10, lines 384-387. We have added the publication to the Reference list as Ref. 46.

Reviewer 2 Report

Comments and Suggestions for Authors

The manuscript biomedicines-2915741 entitled Elevated NET, Calprotectin and Neopterin levels discriminate between disease activity of COVID-19 as evidenced by need of hospitalization for patients in Northern Italy by Geir Hetland , and coworkers aimed to characterize biomarkers that are useful in discriminating the hospitalization risk in a COVID-19 cohort from Northern Italy during the first pandemic wave.

Blood was collected before any anti-inflammatory/immunosuppressive therapy and assessed for soluble C5b-9/C5a, H3-neutrophil extracellular traps (NET), calprotectin, DNase plasma levels by ELISA, and a panel of pro-inflammatory cytokines by ELLA.

Calprotectin and NET levels discriminate between hospitalized and non-hospitalized patients, while DNase negatively correlates with NET levels; there are positive correlations between calprotectin and both NET and neopterin levels.

Neopterin levels increase in patients at the beginning of the disease and more in hospitalized than non-hospitalized ones.

C5a and sC5b-9, and other acute phase proteins correlate with neopterin, calprotectin, and DNase. Both NET and neopterin levels negatively correlate with platelet count.

The research work was well planned and well conducted.

Experimental research is significant.

Figures and tables are informative

Discussion is consistent with results.

Minor revision

Line 136: the producer should be given

Line 178 and following: Figure 1, 2,3 and 4 could be added together in a unique figure with panel A, B, C and D.

Line 233, table 2 should be in one page

A linguistic revision is recommended.

Comments on the Quality of English Language

A linguistic revision is recommended.

Author Response

We would like to thank the Referee for the positive comments.

The manuscript biomedicines-2915741 entitled Elevated NET, Calprotectin and Neopterin levels discriminate between disease activity of COVID-19 as evidenced by need of hospitalization for patients in Northern Italy by Geir Hetland , and coworkers aimed to characterize biomarkers that are useful in discriminating the hospitalization risk in a COVID-19 cohort from Northern Italy during the first pandemic wave. Blood was collected before any anti-inflammatory/immunosuppressive therapy and assessed for soluble C5b-9/C5a, H3-neutrophil extracellular traps (NET), calprotectin, DNase plasma levels by ELISA, and a panel of pro-inflammatory cytokines by ELLA. Calprotectin and NET levels discriminate between hospitalized and non-hospitalized patients, while DNase negatively correlates with NET levels; there are positive correlations between calprotectin and both NET and neopterin levels. Neopterin levels increase in patients at the beginning of the disease and more in hospitalized than non-hospitalized ones. C5a and sC5b-9, and other acute phase proteins correlate with neopterin, calprotectin, and DNase. Both NET and neopterin levels negatively correlate with platelet count.

The research work was well planned and well conducted.

Experimental research is significant.

Figures and tables are informative

Discussion is consistent with results.

Minor revision

Line 136: the producer should be given

We have added the requested information in the “Materials and Methods” section, paragraph “H3-NET dual hybrid ELISA”, page 4, lines 171-174.

Line 178 and following: Figure 1, 2,3 and 4 could be added together in a unique figure with panel A, B, C and D.

As suggested, we have grouped Figures 1, 2 and 3 together as new Figure 1, panels A, B, C. However, we have left Figure 4 (now Figure 2), since neopterin is a parameter of neutrophil functions not strictly relate to NETosis.

Line 233, table 2 should be in one page

Table 2 has been laid out in one page (“Results section”, page 8).

A linguistic revision is recommended.

As suggested, a linguistic revision has been performed.

Reviewer 3 Report

Comments and Suggestions for Authors

In this interesting manuscript authors demonstrate that show that calprotectin, NET, and neopterin are important proinflammatory parameters potentially useful in discriminating between COVID-19 patients at risk for hospitalization. The manuscript is well written and the results well presented. I only have 2 comments prior to be considered for publication in biomedicines:

1) Authors should include more details about statistical analyses performed (in figure legends) and significance (in graphs)

2) Authors should include more details about experiments performed in the materials and methods section

Author Response

We thank the Referee for the constructive suggestions.

In this interesting manuscript authors demonstrate that show that calprotectin, NET, and neopterin are important proinflammatory parameters potentially useful in discriminating between COVID-19 patients at risk for hospitalization. The manuscript is well written and the results well presented. I only have 2 comments prior to be considered for publication in biomedicines:

1) Authors should include more details about statistical analyses performed (in figure legends) and significance (in graphs)

As suggested, we have included additional details on the statistical analysis and its significance in the figures (see the paragraph "Statistics", page 5, line 229, and the legends of the figures 1 and 2). All the changes are highlighted.

2) Authors should include more details about experiments performed in the materials and methods section

We have extended the whole "Materials and Methods" section; see the highlighted changes at pages 3-5.

Round 2

Reviewer 1 Report

Comments and Suggestions for Authors

Thanks for this extremely valuable contribution to the COVID-19 issue.

The uniqueness is associated with an intelligent approach and analysis of a primary innate immune response against a human viral infection.

Reviewer 2 Report

Comments and Suggestions for Authors

the authors emended the manuscript as required improving the quality of the work

Reviewer 3 Report

Comments and Suggestions for Authors

Authors provided all my concerns.